# 3D Reconstruction & Modeling of the Traditional Greek Trechadiri: "Aghia Varvara"

Andreas Arapakopoulos [1,*], Orestis Liaskos [2], Sofia Mitsigkola [2], Georgios Papatzanakis [2], Sofia Peppa [1], Georgios Remoundos [3], Alexandros Ginnis [2], Christos Papadopoulos [2], Dimitrios Mazis [4], Odysseas Tsilikidis [4] and Yannis Yighourtakis [2]

[1] Department of Naval Architecture, University of West Attica, 12243 Athens, Greece; speppa@uniwa.gr
[2] School of Naval Architecture and Marine Engineering, National Technical University of Athens, 10682 Athens, Greece; liaskos.orestis@gmail.com (O.L.); smitsigkola@gmail.com (S.M.); gpap@deslab.ntua.gr (G.P.); ginnis@naval.ntua.gr (A.G.); chpap@central.ntua.gr (C.P.); yiannyiyou@gmail.com (Y.Y.)
[3] Department of Shipping, Trade and Transport, School of Business, University of the Aegean, 82132 Chios, Greece; gremoundos@aegean.gr
[4] Green Maritime Technology Group, 16562 Glyfada, Greece; d.mazis@gmtech.gr (D.M.); o.tsilikidis@gmtech.gr (O.T.)
* Correspondence: andreasarapakopoulos@gmail.com; Tel.: +30-693-490-2277

**Abstract:** 3D modeling techniques have grown increasingly prevalent in a variety of disciplines, including cultural heritage and ship design. The methodology used in the 3D reconstruction of a traditional Greek boat with the Trechadiri hull type named "Aghia Varvara" is presented in this study. The original boat was built in 1925 and is characterized as a modern cultural heritage monument by the Greek Ministry of Culture. The digital reconstruction of the boat is explained in detail, including 3D laser scanning and computer aided geometric design (CAGD), as well as the description of the 3D printing process. The boat's 3D digital model has been used for the enrichment of the NAVS Project's digital library, demonstrating the unique geometrical, typological, and cultural characteristics of Greek traditional shipbuilding, a living craft which listed on Greece's National Inventory of Intangible Cultural Heritage.

**Keywords:** 3D laser scanning; cultural heritage; traditional shipbuilding; point cloud editing; structural design; ship design; 3D printing; exhibition

## 1. Introduction

### 1.1. Related Works

3D modeling and reconstruction of cultural artifacts is becoming increasingly popular, supporting, amongst others, the preservation and promotion of cultural heritage (CH) [1–3]. The use of advanced 3D technologies provides the means of preserving items of cultural heritage [4]. Currently, 3D modelling, 3D visualization, virtual reality, and augmented reality are the most commonly used methods for documenting, recovering and presenting cultural heritage artifacts [5]. The results of 3D modeling can be used to assist and fulfill research efforts on cultural heritage [6].

As far as maritime cultural heritage is concerned, 3D modeling of historical and traditional vessels may be used as an effective and innovative means of depicting and highlighting maritime history [7–9] and traditional shipbuilding [10].

Different techniques have been employed to carry out the digital documentation of ships, but there is no standard [11]. Many research studies used laser scanning to document traditional ships, including those of [12–14].

Cultural institutions and museums use new media to engage with new audiences and provide new ways to impart historical facts, expanding the reach of knowledge to

visitors [15–17]. Furthermore, 3D printing technology can be used to recreate and preserve historical and traditional artifacts [18–20].

### 1.2. Greek Traditional Shipbuilding & Trechadiri

Wooden shipbuilding (or traditional shipbuilding) is the handcrafted construction of a vessel using natural wood [21] involving the vessel's conception, design, construction, equipment, and ornamentation, as well as the associated cultural activities, social practices and values [22–24]. In 2013, the National Inventory of Intangible Cultural Heritage of Greece added wooden shipbuilding (or traditional shipbuilding) to its list of intangible cultural heritage [25].

Trechadiri is one of the most known traditional Greek vessels. The first evidence of Trechadiri boat building is from the mid-seventeenth century. Trechadiri was first constructed in 1658 on the Greek island of Hydra. This boat type was common in the 19th century [26] in the islands of the Argosaronic Gulf. Hydra and Spetses were the islands where the majority of the Trechadiri crafts were erected in the 18th and 19th centuries. Boats with a Trechadiri hull type were built in nearly every small shipyard of the Aegean Sea in the beginning of the twentieth century. Most Trechadiri hull boats today have an overall length of 8 to 20 m and a carrying capacity of 4 to 50 tons. The concave curving shape of the stem post is the most distinguishing feature of the hull of a Trechadiri. Another characteristic feature is that the stern post is raked aft and is practically straight. Older drawings, on the other hand, depict the stern post as having a gentle curve. A smooth curve of the hull at the bilge level characterizes the middle section of a Trechadiri. The two upper sides of this section are shaped on a bow and buttock plan with an angle of around 15 degrees [22–24].

### 1.3. "Aghia Varvara"

"Aghia Varvara" is a former fishing boat which was declassified and withdrawn about 20 years ago, turning into a museum exhibit, within the framework of the Community legislation for the reduction of overfishing [27]. The boat has a corresponding certificate from the Ministry of Rural Development [28] and is owned by the Municipality of Perama (hosted by Maritime Tradition Museum [29] since 2011). The general characteristics of this boat are the following:

**Type:** Trechadiri,
**Place of Construction:** Perama, Greece,
**Year of Construction**: 1925
**Length Overall**: 10.60 m, **Beam:** 4.55 m

In Figure 1, a perspective view of the Trechadiri "Aghia Varvara" in its present state is illustrated. Figure 2 also depicts the boat's deck as well as its appendages (propeller and rudder). A close examination of the hull reveals multiple cracks in the deck timber and the boat's stern.

The internal structure of the hull where the frames and longitudinal strengthening components are shown in Figure 3. Many unnecessary objects are observed, which are not generated during the reconstruction process.

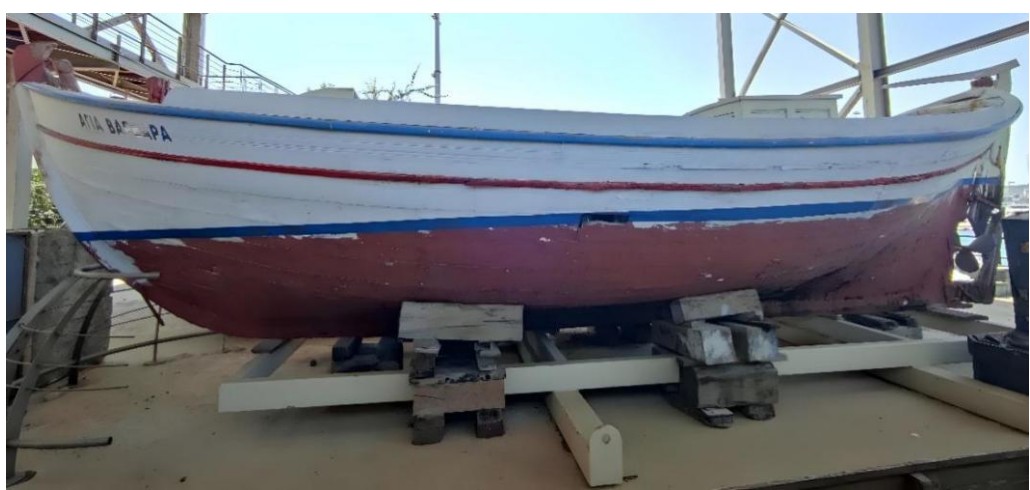

**Figure 1.** The Greek Trechadiri "Aghia Varvara" (perspective view).

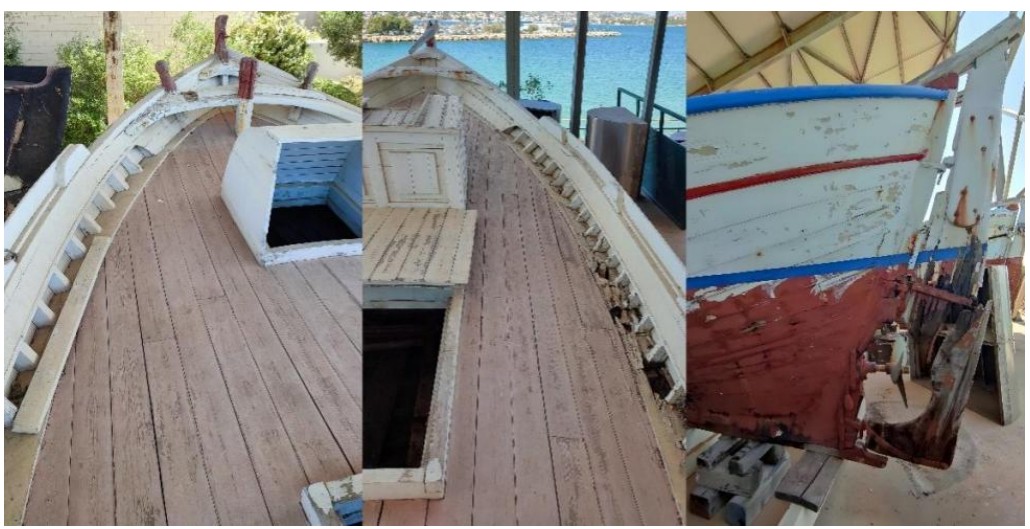

**Figure 2.** Deck of the "Agia Varvara" (**left**/middle photos); stern side with propeller and rudder (**right** photo).

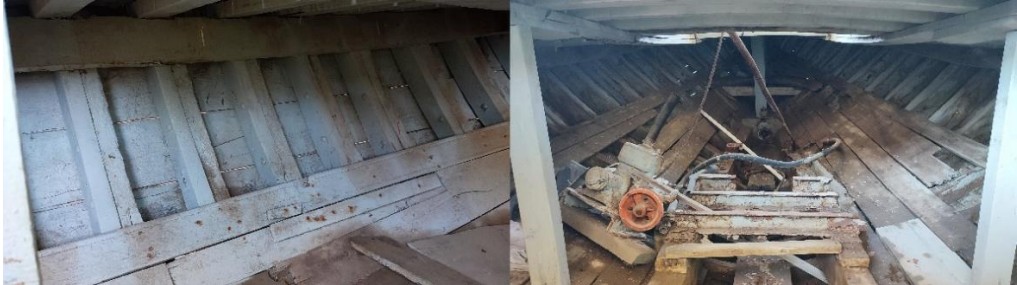

**Figure 3.** Internal structure of the hull: frame sections and longitudinal strengthening components (**right**, **left** photos).

### 1.4. Objectives

This paper presents the implementation of consolidated methods and procedures for 3D recreation of the hull of a traditional boat. The actions, decisions, and methodology of the reconstruction process are all described and analyzed in detail. The reconstruction of "Aghia Varvara" aims toward the creation of a digital model and a printed model under scale. The models are created to show the boat in its original shape, ignoring any existing

damage, and providing as much comprehensive and precise design information as possible to highlight the unique hull type and retain traditional shipbuilding's cultural features. This is significant since traditional shipbuilding is not documented, as ships were 'built by eye' using 'rules-of-thumb' [30], and most hulls are only partially preserved, lacking major structural elements, as a result of deformation processes.

The research's main goals and objectives are as follows:

- to analyze and present the digital model reconstruction techniques using 3D technology: 3D laser scanning and editing of point clouds, Rhino3D Design, texturing, and 3D printing
- to contribute to culture preservation by recreating a digital model of a Trechadiri type boat with as much detailed and precise design information as feasible.

The results of this research may be useful to researchers in cases of developing digital databases for determining traditional hull types and improving 3D modeling to be utilized in digital libraries and virtual museums.

## 2. 3D Reconstruction Process

### 2.1. 3D Laser Scanning

3D laser scanning is a non-contact, non-destructive technology that digitally captures the shape of physical objects using a line of laser light. 3D laser scanners create "point clouds" [31] of data from the surface of an object. In other words, 3D laser scanning is a way to capture a physical object's exact size and shape into the computer world as a digital 3D representation.

A state-of-the-art 3D laser scanner has been used by the G.M.T Group [32] engineers during the present onboard survey. The 3D laser scanner (Faro Focus S70, Lake Mary, FL, USA) [33] is shown on the left of Figure 4, spherical targets are used to register the spot scans, and are shown on the middle of Figure 4. Further, checkerboard targets (shown on the right of Figure 4) are also used as additional targets, mainly for plane surfaces, to ensure the overall accuracy of the registration process.

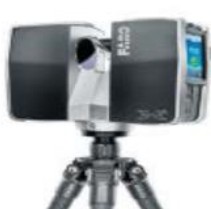 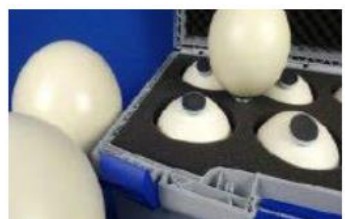 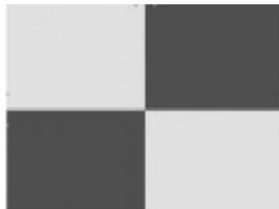

**Figure 4. Left**—FARO Laser scanner Focus 3D; **Middle**—Sphere targets; **Right**—Checkboard target.

The 3D laser scan procedure was carried out as part of the NAVS project on the grounds of the Maritime Tradition Museum [29] under the consent and kind support of its administration. Vibration and noise issues have not influenced the scanned data. Other barriers in the area, such as the boat's supports in Figure 5, could have altered the scanned data. However, the point cloud of "Aghia Varvara" has millimeter-level accuracy.

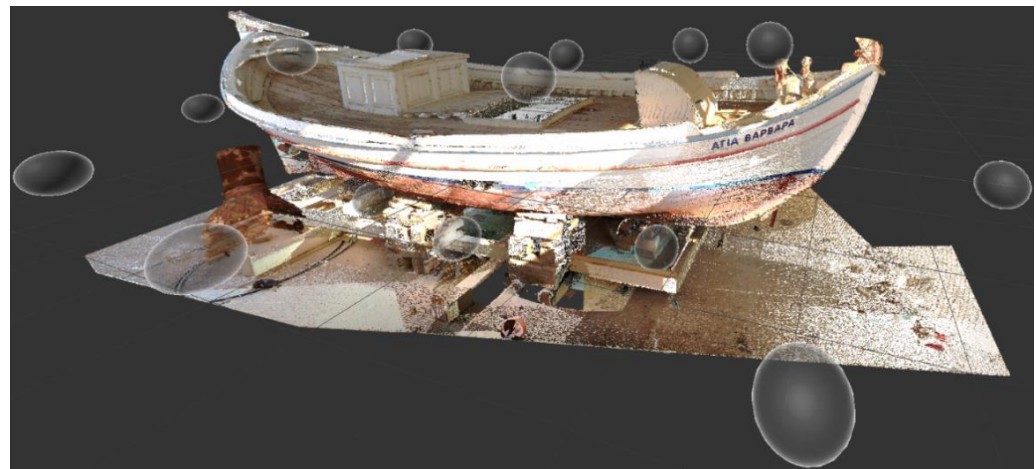

**Figure 5.** Scanned boat-model in Autodesk ReCap.

Upon 3D scan survey completion [34], Autodesk ReCap [35] was used in order to process data and produce the final registration files in which all scanned areas are depicted in a point cloud format for further use and post-processing. In Figure 5, a representative point cloud produced is presented in Autodesk ReCap concerning the areas of interest of the particular vessel.

The bubbles, which are depicted in Figure 5, correspond to the spots in which a scan has been done. The total number of scan positions is 26.

In the next step, the unnecessary objects from the surrounding area around the boat are erased (Figure 6) and a .e57 [36] file is exported from Autodesk ReCap and imported to Rhino3D [24], as also shown in Figure 6.

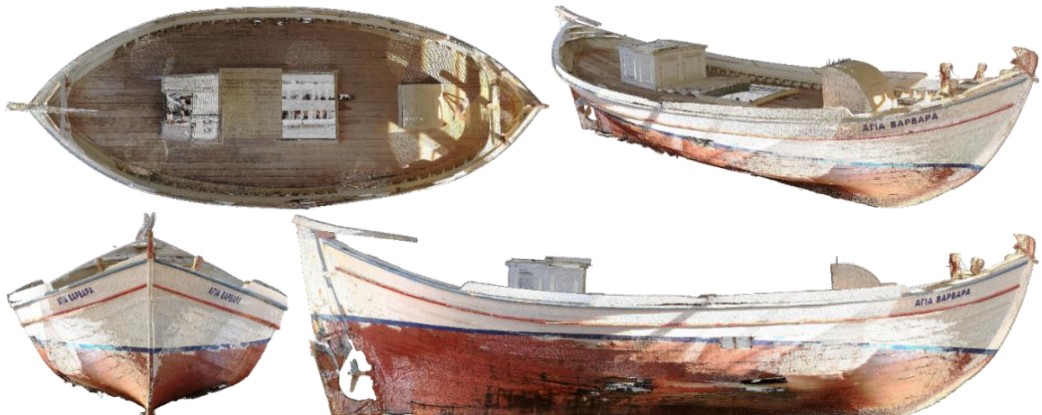

**Figure 6.** Point cloud from the laser scanner show the image of the Trechadiri: Four views of the boat imported in Rhino3D.

### 2.2. Design in Rhino3D

The design in Rhino3D [37] initiates by processing the imported point cloud (Figure 6). The outcome of this process is red interpolation curves, which are shown at the top of Figure 7. Afterwards, green smooth NURBS [38,39] curves are generated from the red ones to represent the ship sections. These are the control curves of the outer hull surface, which is depicted at the bottom of Figure 7. Following the information of the point cloud and the hull surface a solid (closed polysurface) is generated together with proper openings, as seen at the bottom of Figure 7.

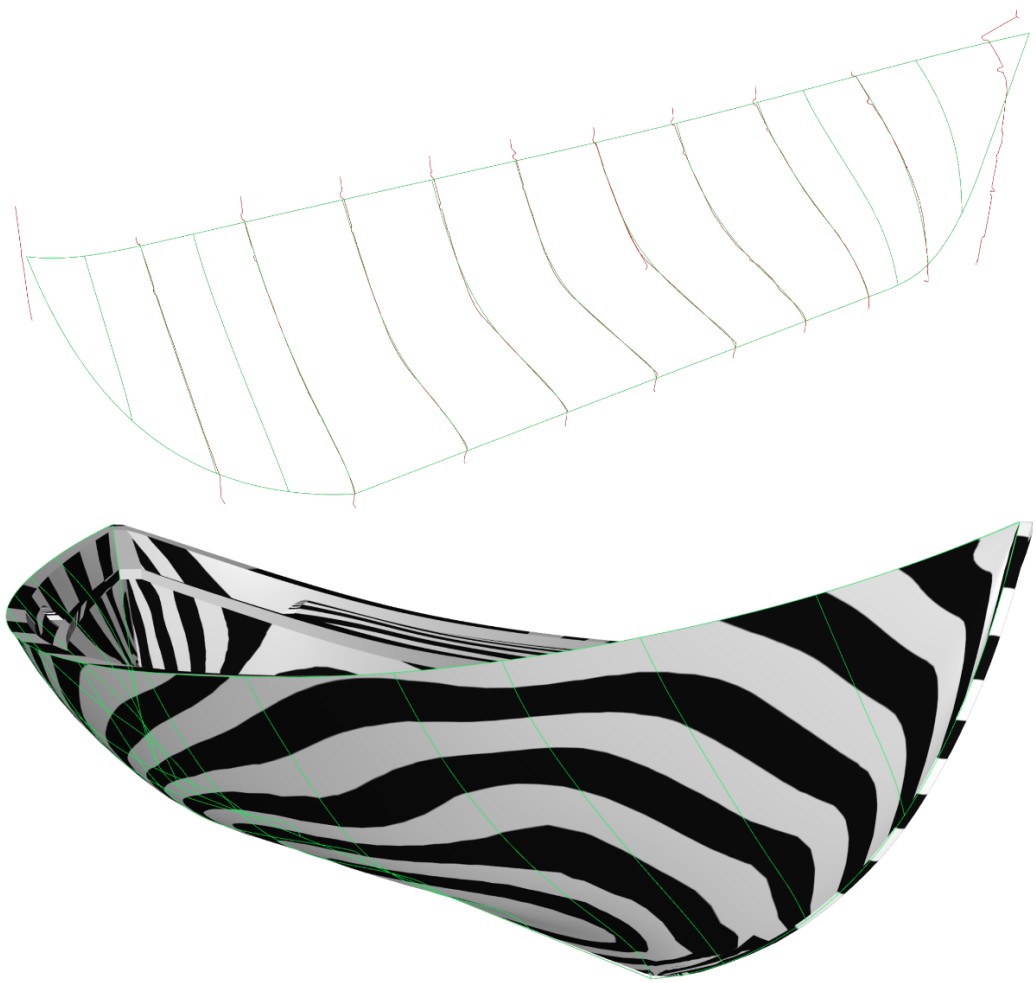

**Figure 7.** Hull design in Rhino3D: ship sections (**top**); hull surfaces & solid (**bottom**).

In addition, Figure 7 (bottom) shows a visual surface analysis using the Rhino3d Zebra command [40] to highlight the smoothness and continuity of the entire closed polysurface. The hull surface indicates G2 (position + tangency + curvature) continuity on the entire surface. At the connection between the hull surface and the adjacent surfaces G0 (position) continuity is indicated.

Based on the aforementioned visual surface, the transverse stiffeners were designed, as depicted in green and red color. The differentiation on the color indicates the existence of two different wooden strips. The red ones are longer and visible above the deck. Both red and green stiffeners end on the keel with blue color, as shown in Figure 8, and they correspond to the pair of sections at the top of Figure 3.

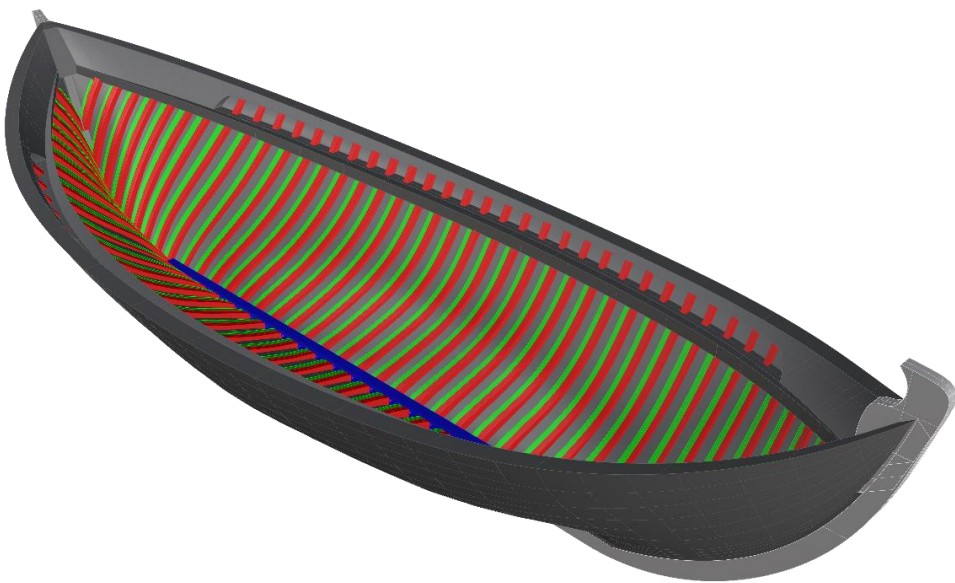

**Figure 8.** Pair of structural sections.

In Figure 9 the cyan colored wooden strips are depicted. These strips support the entire deck and connect the transverse stiffeners for extra durability. The vertical stiffeners were also colored cyan. The purple and the gold wooden strips support the transverse stiffeners. The colors' variation represents the different kind of stiffeners supporting the boat's structure.

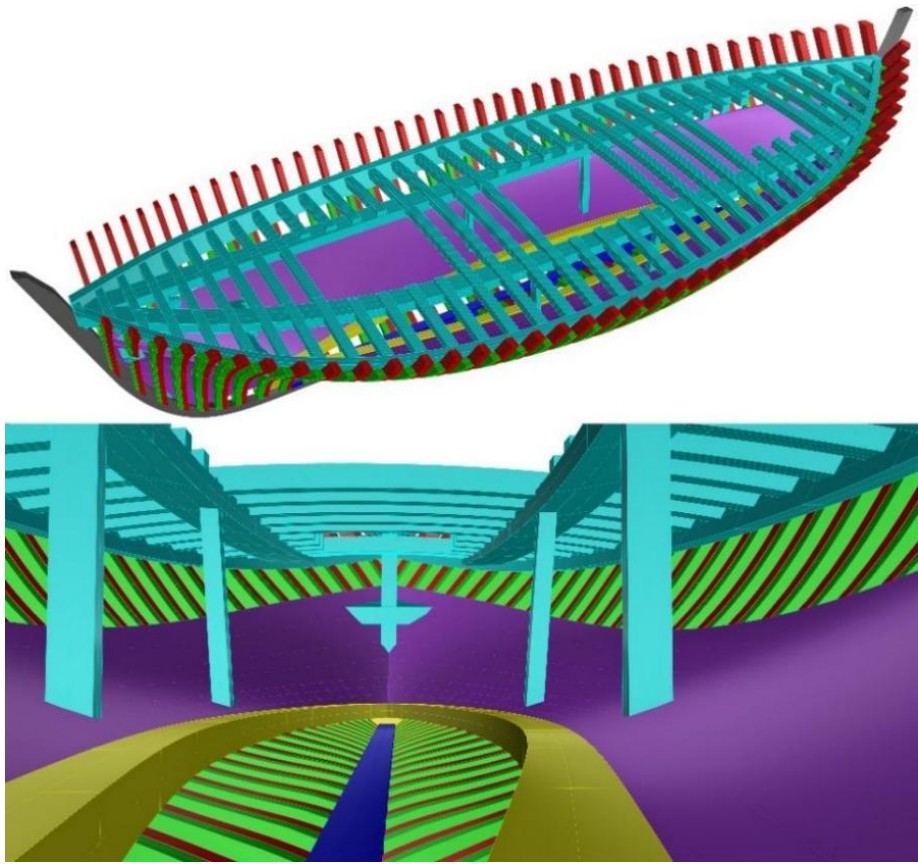

**Figure 9.** Hull design in Rhino3D: skeleton (**top**); details of inner area (**bottom**).

The final model together with the rudder, the propeller and the deck parts designed in Rhino3D is appeared in Figure 10. Due to lack of information about the model of the propeller, a Wageningen B3-60 [41] was chosen (as a conventional approach to the unknown original one) and it was designed by the parametric modeler presented on [42].

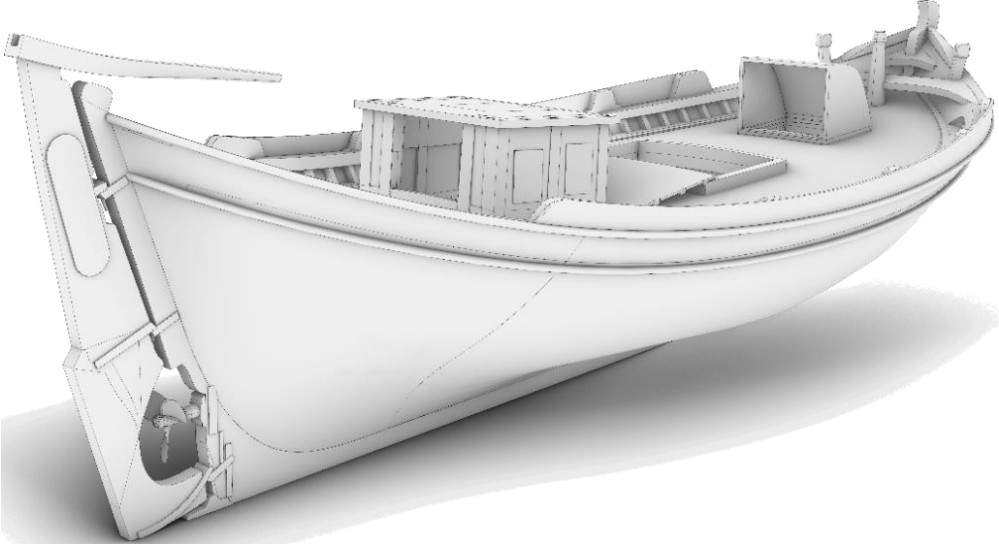

**Figure 10.** 3D model of "Aghia Varvara".

Figure 11 shows the small deviation between the 3D model and the point cloud. A visual comparison can also be done between the original boat (Figures 1–3) and the 3D model (Figures 8–10).

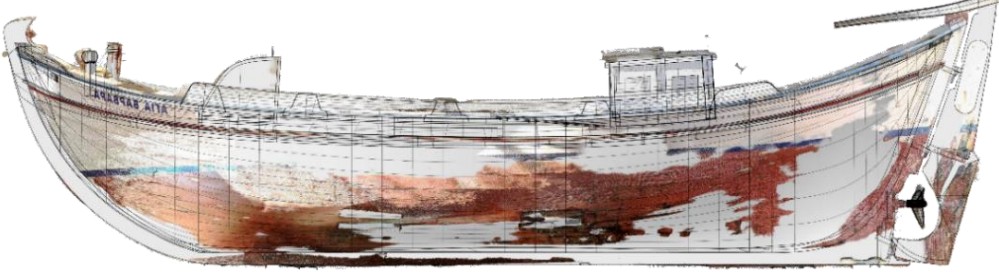

**Figure 11.** Comparison between 3D model and point cloud.

Figure 12 is a rendering of the boat without its hull, to further illustrate the complexity of its stiffener arrangement.

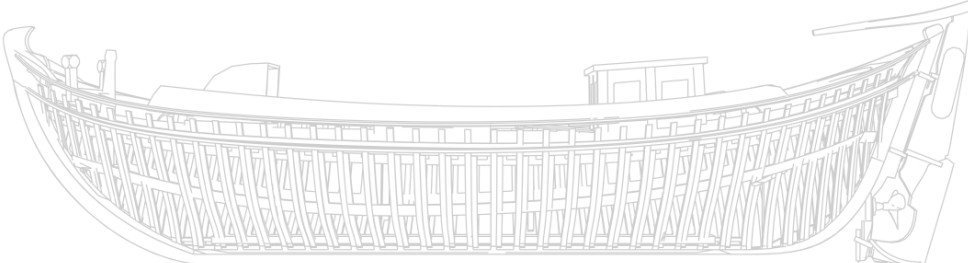

**Figure 12.** Rendering of the "Aghia Varvara" without its hull.

Once the solid is completed, it needs to be retopologized [43]. Rhino3D offers a built-in tool for such purpose, QuadRemesh [44], which produces a mesh that is useful in the following step (texturing) since the generated topology is rather simple, as shown in Figure 13.

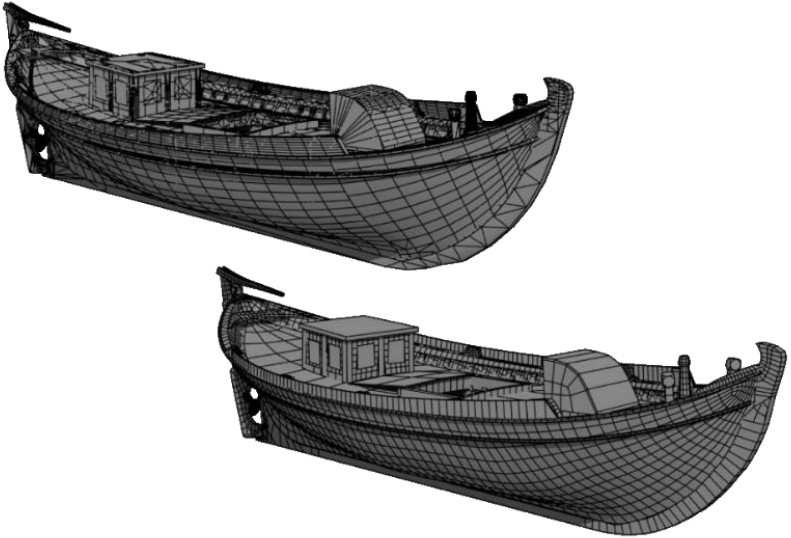

**Figure 13.** 3D model before QuadRemesh (**Top**); 3D model after QuadRemesh (**Bottom**).

*2.3. Texturing*

After "Aghia Varvara" was retopologized (Figure 13) and UV unwrapped using Blender [45], detailed maps were created with the use of Adobe Substance 3D Painter [46]. The process followed for this is similar to the one followed for "The Ships of Navarino", although "Aghia Varvara" is much less demanding in materials, and due to the fact that this is not a Virtual Reality project reducing the texture quality is not deemed necessary [47]. Reference pictures taken from the existing boat were used in the creation of texture maps (Figure 14). A matter of historical significance is the positioning of the waterline that needs to be hand painted in order to reassure resemblance to the actual boat.

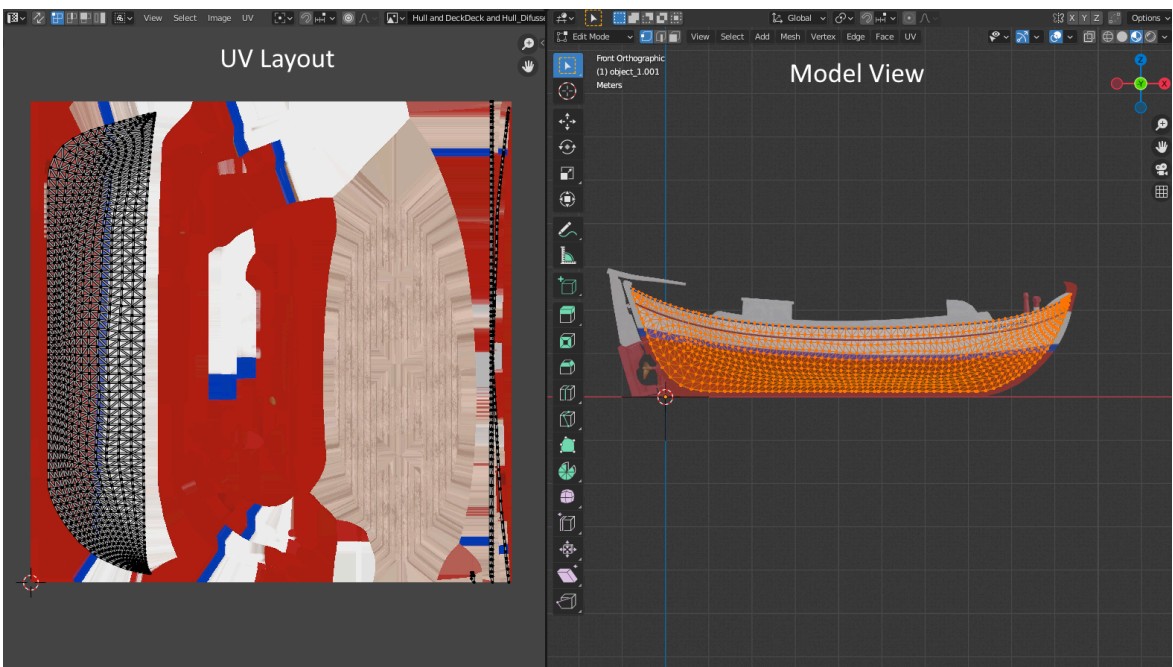

**Figure 14.** "Aghia Varvara" 3D model: UV Layout (**left**); Model View in Blender (**right**).

## 3. 3D Rendering & 3D Printing

After the 3D textured model has been generated, the Cycles [48] render engine of Blender was used to apply the 3D rendering process. 3D rendering is a computer graphics technique that converts a digital three-dimensional scene into a two-dimensional image. The developed photorealistic images of "Aghia Varvara" with proper lightning are shown in Figures 15 and 16.

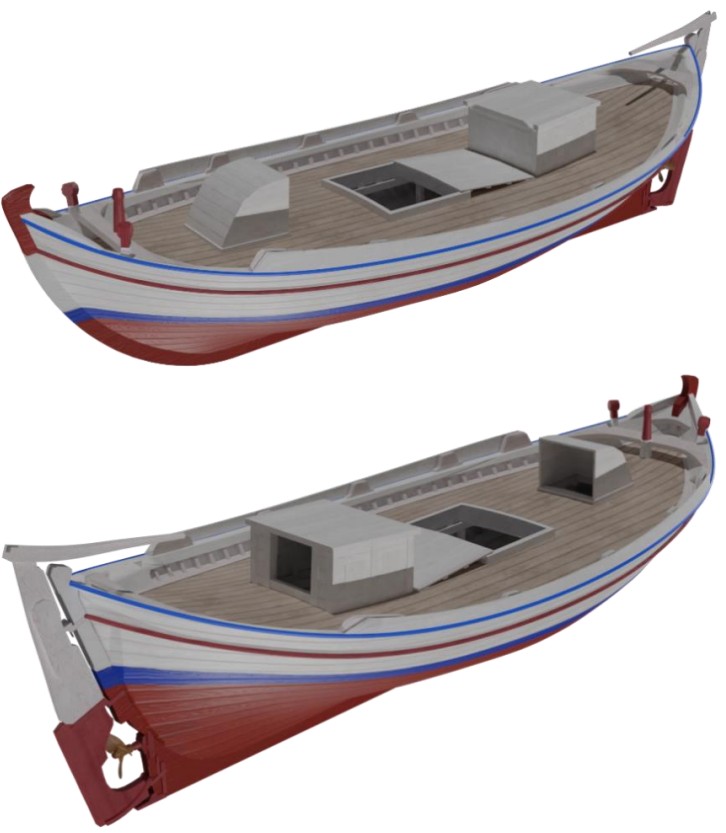

**Figure 15.** Two views of the rendered model of "Aghia Varvara" in Blender.

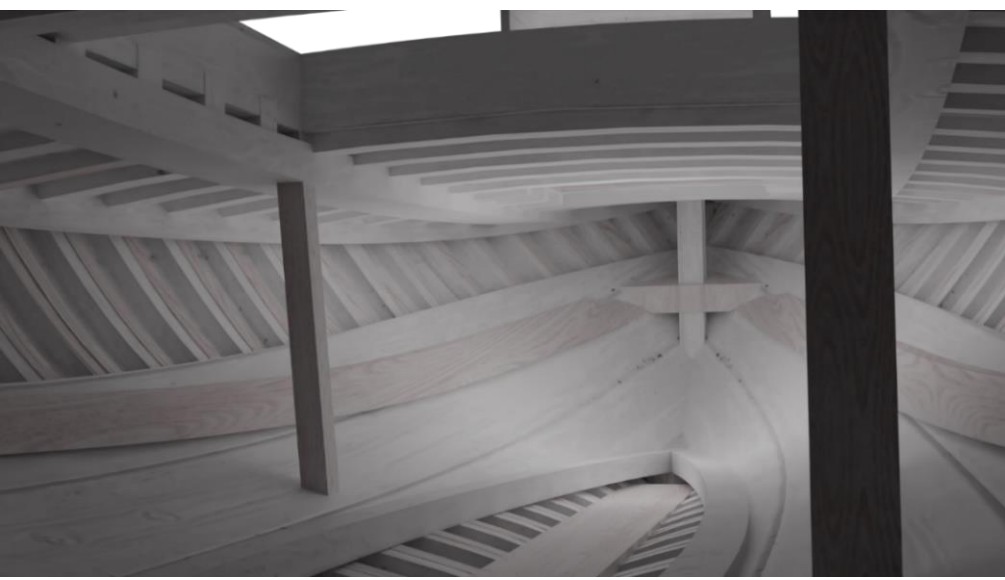

**Figure 16.** Rendered details of inner area in Blender.

In particular, Figure 15 depicts two perspective views of the final rendered model in Blender, while Figure 16 depicts the inner area of the boat, illustrating the completed 3D model. Once again, a visual comparison between the rendered model and the real boat is provided (Figures 1–3).

Furthermore, the ship model was printed on an SLA [49] 3D printer named iSLA300 [50]. Generally, the printing process in the case of "Aghia Varvara" is similar with the one followed for "The Ships of Navarino" [47], since the same printer was used in both cases.

There is a slight increment on the scaling ratio (1:25) of the printed model compared to the one used for printing the models of "The Ships of Navarino" (1:125), due to the fact that there is a rather large difference between the overall length of the respective ships.

Due to the printing volume constraints of the specific printer (300 mm × 300 mm × 300 mm), the hull was printed into parts in order to generate the model as large as feasible, while part of the hull was omitted, as illustrated in Figures 17 and 18, facilitating, in that way, the visibility for the visitors of the exhibition.

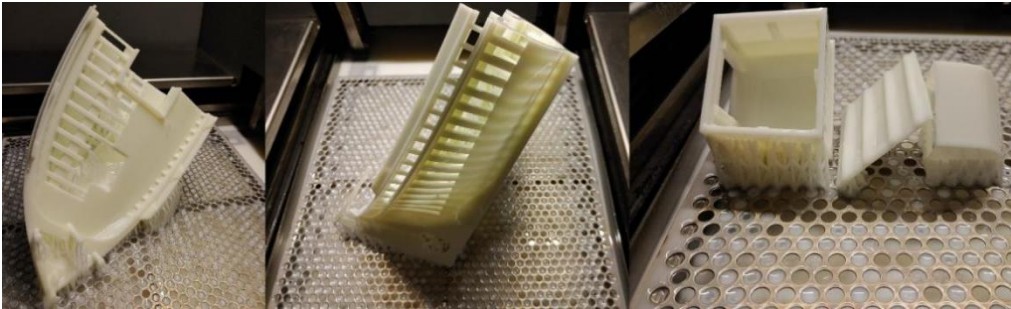

**Figure 17.** Hull parts during the printing process: bow (**left**); stern (**middle**); deck parts (**right**).

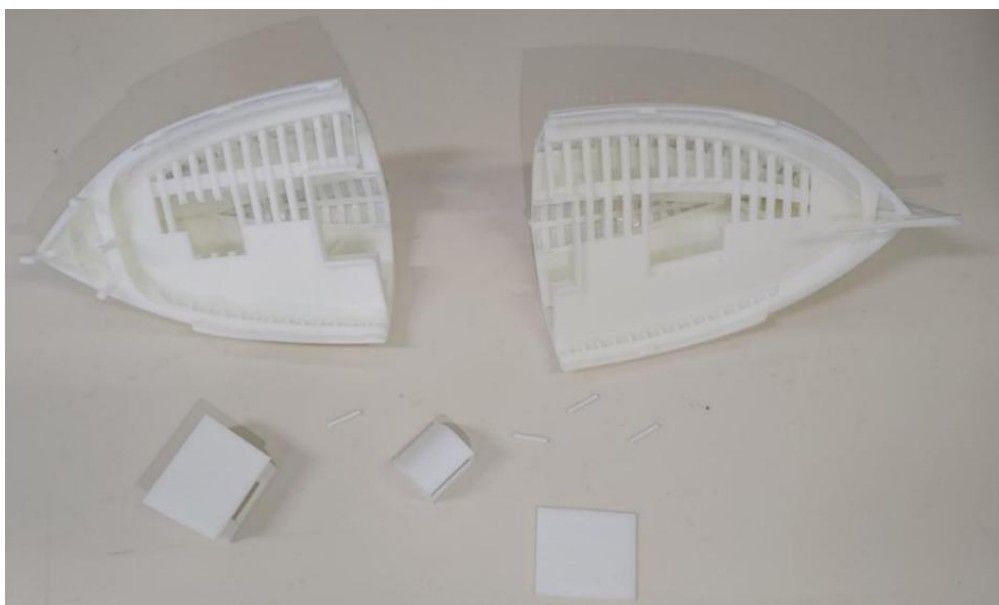

**Figure 18.** Finished products after alcohol rinse and UV cure.

During the printing process supports on the printed parts were generated for better stability, as shown in Figure 17 and later they were removed (Figure 18).

In addition, the printed parts were brushed and rinsed in isopropyl alcohol (IPA) and afterwards they were placed in a UV Curing Chamber [51] to improve their hardness and stability. The SLA parts were painted and then assembled by an artist, as shown in Figure 19.

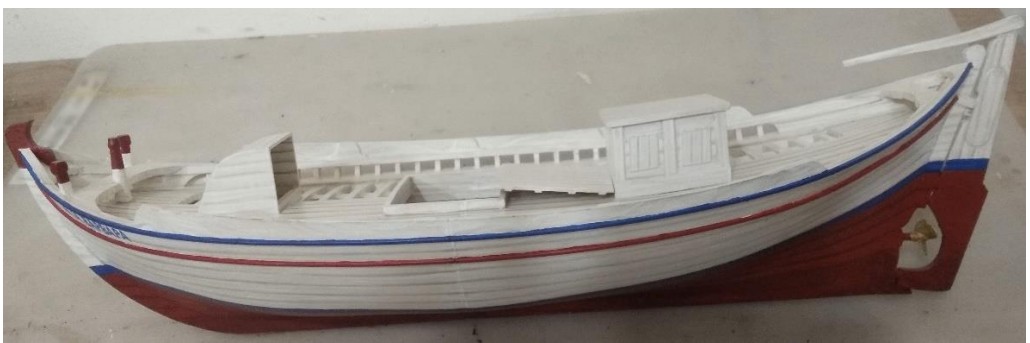

**Figure 19.** 3D printed model of "Aghia Varvara" after the painting process.

## 4. Discussion

This paper describes how the 3D laser scanning, CAD, texturing, and 3D printing may be combined to recreate a historical and traditional boat, such as the Trechadiri "Aghia Varvara". The entire 3D reconstruction process was presented in detail. In addition, an accurate digital model of the "Aghia Varvara" with 3D geometry was created and a physical model of the boat was produced using a 3D printer.

This work was carried out as part of the NAVS Project and the 3D geometric of "Aghia Varvara" vessel was used for the development of the digital database of geometric ship models for the NAVS Project accompanied by digital documentation (3D designs, point clouds), photographs, and historical information [52]. One of the project's goals was to digitally capture designs of traditional boat hull types, such as the Trechadiri hull type, contributing toward the preservation of Greek wooden shipbuilding heritage. Also, the 3D-printed model of "Aghia Varvara" was painted in the original boat's colors in order to be presented in an exhibition titled "The NAVS Project: a novel way to study and preserve traditional shipbuilding heritage" hosted at the Eugenides Foundation in Athens, Greece.

Furthermore, keeping in mind that virtual and augmented reality may provide a more immersive experience, digital ship models generated according to the presented process, such as the one of "Aghia Varvara", might be used in the development of a virtual museum in which visitors have the opportunity to explore the distinctive geometry of the hull types of the traditional Greek boats, online, via a WebGl application [53].

**Author Contributions:** Conceptualization, O.L., S.M., A.A., A.G., C.P. and S.P.; Data curation, O.L., S.M., A.A., A.G., C.P. and S.P.; Funding acquisition, A.G., C.P. and G.R.; Investigation, O.L., S.M., A.A., S.P. and G.R.; Methodology, O.L., S.M., G.P. and S.P.; Project administration, O.L., A.A., A.G. and C.P.; Resources, O.L., S.M., A.A., G.P., A.G. and C.P.; Software, O.L., G.P., A.G., D.M. and O.T.; Supervision, O.L., G.R. and S.P.; Validation, S.M. and G.P.; Visualization, O.L., S.M., A.A., G.P. and Y.Y.; Writing—original draft, O.L., S.M., A.A., S.P. and G.R.; Writing—review & editing, O.L., S.M., A.A., G.P., A.G., C.P., S.P., G.R., D.M., O.T. and Y.Y. All authors have read and agreed to the published version of the manuscript.

**Funding:** The NAVS Project—promotion, documentation, and technical support of "The Greek shipbuilding legacy—the Battles of Navarino and Salamis" launched in June 2020, co-financed by the European Regional Development Fund of the European Union and Greek national funds through the Operational Program Competitiveness, Entrepreneurship and Innovation, under the call RESEARCH–CREATE–INNOVATE (project code: T1EDK-05103).

**Institutional Review Board Statement:** Not applicable.

**Informed Consent Statement:** Not applicable.

**Data Availability Statement:** Not applicable.

**Acknowledgments:** The authors would like to thank all those who participated in the project. In particular, Green Maritime Technology (GMT) who contributed to the 3D laser scanning of the ship model. Additionally, the team would like to thank E. Nomikou and C. Troumpetari from the Eugenides Foundation for commenting on drafts of this paper, as well as associate researcher A. Pagonas for his support on the 3D reconstruction & modeling of "Aghia Varvara".

**Conflicts of Interest:** The authors declare no conflict of interest.

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
