# Peer review of "3D Reconstruction & Modeling of the Traditional Greek Trechadiri: “Aghia Varvara”"

_heritage, doi:10.3390/heritage5020067_

Round 1

Reviewer 1 Report

The paper describes an application of consolidated methods and procedures of digital survey applied to Cultural Heritage. The different steps are well described. So the main goal of this research (3D recreation of a traditional Greek boat) has been reached. However from a scientific point of view it could be interesting to insert some critical analysis related to the study of the hull shape, analyzing i.e the geometric properties of the NURBS surface obtained from points cloud to highlight those specific characteristic of a Trechadiri vessel. 
The are a lot of study about the analysis of a hull shape. In this way the paper could be surely more interesting and original for the scientific community.

Author Response

Please, check the attached document file.

Reviewer 2 Report

The work is very short, without great conceptual developments or state of the art. It doesn't discusses future research devlopment and the discussion topics are not well addressed. I would like to have more about how innovative this contribution is and what can bring to future research and heritge safeguard and knowledge transmission. 

Author Response

(The authors gave the same response as above.)

Reviewer 3 Report

In this paper, the authors present the 3D model of the "Aghia Varvara" to feature the unique geometrical, typological and cultural characteristics of Greek traditional shipbuilding. For this, 3D laser scanning, Computer-Aided Geometric Design (CAGD), texturing and 3D printing were used in the 3D reconstruction of the model.

This work was carried out as part of the NAVS project - Promotion, documentation, and technical support of “The Greek shipbuilding legacy—the Battles of Navarino and Salamis” launched in June 2020, co-financed by the European Regional Development Fund of the European Union and Greek national funds through the Operational Program Competitiveness, Entrepreneurship and Innovation, under the call RESEARCH–CREATE–INNOVATE (project code: T1EDK-05103).

This work to recover ancient historical heritage seems interesting and the methodology is well-known (accurate digital model by 3D laser scanning). However, I have some additional comments:

First, the section 1.1. Related works is very poor. It should be improved with some current references related to the ship design (previous experiences from other authors related to the ancient maritime cultural heritage) including their main findings and drawbacks.  

On the other hand, it is necessary to include a new sub-section (1.4. Objectives), to clearly state the objectives of this research and to express the possible additional uses of the digital model to be obtained (for example, its use in the virtual and augmented reality, WebGL models, diffusion through the network, digital museums, the possibility of creating bookstores of reconstructed historical heritage, etc…)

Likewise, the design process with Rhino3D is well explained and detailed with quality images, but it would be convenient for the authors to explain why they have used Autodesk Recap for the treatment of the point cloud obtained after the use of the 3D laser scanner and its subsequent refinement, compared to other commercial software (indicating the main advantages and disadvantages). Similarly, explain the use of Rhino3D in obtaining the final accurate digital model, compared to other commercial software. 

Finally, it would be interesting that the authors present the future developments following this line of research.

Regarding formal aspects:

1.        Please, improve the abstract including only the objectives, methodology, main results and conclusions.

2.        Please, check the grammar and spelling of the English.

Author Response

(The authors gave the same response as above.)

Round 2

Reviewer 2 Report

There are a few observations in the attached file because the modification made really resulted in much better text. 

In general, I would like to add that this article could be improved on the analysis of results and contribution to society. Some subjects could be better integrated between sections.

Author Response

This manuscript is a resubmission of an earlier submission. The following is a list of the peer review reports and author responses from that submission.